# A New Fault Diagnosis of Rolling Bearing Based on Markov Transition Field and CNN

**DOI:** 10.3390/e24060751

**Published:** 2022-05-25

**Authors:** Mengjiao Wang, Wenjie Wang, Xinan Zhang, Herbert Ho-Ching Iu

**Affiliations:** 1School of Automation and Electronic Information, Xiangtan University, Xiangtan 411105, China; wwj291968051@163.com; 2School of Electrical, Electronic and Computer Engineering, University of Western Australia, Crawley, Perth, WA 6009, Australia; xinan.zhang@uwa.edu.au (X.Z.); herbert.iu@uwa.edu.au (H.H.-C.I.)

**Keywords:** feature extraction, Markov transition field, convolutional neural network, fault diagnosis, rolling bearing

## Abstract

The rolling bearing is a crucial component of the rotating machine, and it is particularly vital to ensure its normal operation. In addition, the selection of different category features will add uncertainty and bias to the classification results. In order to decrease the interference of these factors to fault diagnosis, a new method that automatically learns the features of the data combined with Markov transition field (MTF) and convolutional neural network (CNN) is proposed in this paper, namely MTF-CNN. The MTF contributes to convert the original time series into corresponding figures, and the CNN is used to extract the deep feature information in the figure to complete the fault diagnosis. The effectiveness of the proposed method is verified by two public data sets. The experimental results show that MTF-CNN can classify different types of faults, and the highest accuracy rate can reach 100%. Likewise, the classification accuracy of this method is higher than some existing methods.

## 1. Introduction

In the modern industrial system, rotating machinery has been widely used, so it is particularly important to ensure the stable operation of machinery and equipment [1,2]. Rolling bearings are a crucial part of rotating machinery. Due to the influence of its work environment and other factors, rolling bearings have a higher probability of failure than other components [3,4,5]. Hence, the accurate detection of rolling bearing faults has great practical value.

With the advancement of sensor technology and signal processing methods, detecting equipment failures through the vibration signals of rotating machines becomes increasingly popular [6,7,8]. The signal processing method can effectively extract the characteristics of the original vibration signal in the field domain [9,10,11]. The signal processing methods mainly include: wavelet transformation (WT) [12], local mean decomposition (LMD) [13], empirical mode decomposition (EMD) [14], variational mode decomposition (VMD) [15] and so on. After extracting the hidden features form the time series, the full fault diagnosis process is completed by the combinational usage of several artificial intelligence methods, such as random forest (RF) [16], support vector machines (SVM) [17] and k-nearest neighbor (kNN) [18]. Zheng et al. achieved fault classification by calculating the composite multiscale fuzzy entropy (CMFE) of the vibration signal based on ensemble support vector machines (ESVM) [19]. Yan et al. decompose the vibration signal through an improved VMD (IVMD) to obtain the components. They proposed instantaneous energy distribution-permutation entropy (IED-PE) to obtain the feature vectors. At the same time, combined with kNN, the fault diagnosis of rolling bearing is realized [20]; however, most of the traditional processing methods count on manually extracting the fault information from the vibration signal, which requires tedious signal processing work. In fact, the expertise of people who carry out signal processing can greatly impact the quality of classification results.

Currently, with the rapid progress of mathematical theory and computer technology, deep learning (DL) has been widely used. DL can automatically extract the non-linear relationships between variables from the data, avoiding the manual extraction process [21]; therefore, it is feasible to judge the running state of rolling bearings by the DL method. Such as deep belief network (DBN) [22], deep neural network (DNN) [23], sparse auto-encoder [24], convolutional neural network (CNN) [25] and so on. CNN, as one of the most popular DL methods, can effectively extract deep features of data. Wang et al. enhanced the relevant characteristics of the fault and realized the classification of the fault with 1D-CNN [26]. Compared with the other DL methods, CNN has obvious advantages in processing pictures. The CNN model is specially designed for processing 2-D or 3-D data. In some studies, the raw data are converted into pictures, and then the methods of image processing are used to classify the pictures to complete the entire process. Li et al. predicted the rest life of the bearing by extracting the time–frequency information of the data sequence and using CNN to complete feature extraction [27]. Wen et al. realized fault diagnosis on multiple platforms such as rolling bearings by converting original vibration signals into figures and combining with CNN [28].

The main contributions of this paper are as follows. Firstly, a new method of converting time series data into figures using the Markov transformation field (MTF) [29] is introduced. MTF is based on the Markov transition matrix, which is obtained from the first-order Markov chain. Due to the memorylessness of the Markov chain, the current elements in the Markov matrix are independent of the previous element, which results in the loss of much information in the conversion process. By introducing the time axis, more time information can be retained during the conversion process; therefore, the MTF transforms time series into corresponding images, which can decrease information loss. Then, via the CNN model applied in this paper, by reducing the level of convolutional layers and optimizing the parameters of the fully connected layer, the operating efficiency is improved. Finally, by combining the MTF with the CNN model proposed (MTF-CNN), a new method for fault diagnosis of rolling bearings is proposed.

The rest sections of this paper are organized as follows. Section 2 introduces the theory of MTF. Section 3 is the background of CNN. Section 4 describes the fault diagnosis method MTF-CNN and the structure of the CNN adopted. Section 5 verifies the effectiveness of MTF-CNN and compared with other DL methods in accuracy. Finally, the conclusions are drawn in Section 6

## 2. MTF Theory

### 2.1. State Transition Probability Matrix

State refers to the current system status. For event *S*, when we assume that there are *n* different states during its development, and the *i*-th state in the *t* moment is $S_{t}$. Then, the development process of events can be represented by the following:(1)$$S=\{S_{1},S_{2},\cdots ,S_{n}\}$$

The probabilities of reaching a state is expressed as $E_{i}\left(t\right)$.
(2)$$E_{i}\left(t\right)=p\left\{X_{t}=S_{t}\right\},t=1,2,\cdots ,n$$

Therefore, the state space of the research object includes *k* parts as expressed in (Equation 3) below.
(3)$$T_{k}=\left\{E_{1}\left(k\right),E_{2}\left(k\right),\cdots ,E_{n}\left(k\right)\right\}$$
according to the probability theory, $E_{i}\left(k\right)$ satisfies the condition: $\overset{n}{\underset{i=1}{\sum }}E_{i}\left(k\right)=1,E_{i}\left(k\right)\geq 0$.

The state of the research object will change over time. When the state of the research object transform, $S_{i}$ to $S_{j}$, this state change is probabilistic, which is called state transition probability. According to the number of steps of state transition, the state transition probability can be divided into two types: one-step and multi-step.

One-step transition probability refers to the probability of changing one state to the other state. The research subject is in $S_{i}$ at t=T, and the probability of state change $S_{j}$ at t=T+1 is $P_{ij}$. Its expression is as follows:(4)$$P_{ij}=p\left\{S_{i}|S_{j}\right\}=p\left\{X_{T+1}=S_{j}|X_{T}=S_{i}\right\}$$

Multi-step transition probability indicates the probability of continuous atate changing for multiple steps. The research subject is in $S_{i}$ at t=T, after *k* times of state changes, and the probability of state change $S_{j}$ at t=T+1 is $P_{ij}$. Its expression is as follows:(5)$$P_{ij}=p\left\{S_{j}|S_{i}\right\}=p\left\{X_{T}=S_{j}|X_{T+k}=S_{i}\right\},k=2,3,\cdots$$

During the entire transition process, the range of the state transition probability value is (0,1) and satisfies the following relationship:(6)$$\sum_{i,j=1}^{n}P_{ij}=1,\left(i,j=1,2,\cdots ,n\right)$$

The matrix composed of these transformation probabilities is called the transformation probability matrix, and the expression is as follows:(7)$$P={\left(P_{ij}\right)}_{N\times N}=\left[\begin{array}{cccc} P_{11} & P_{12} & \cdots & P_{1n}\\ P_{21} & P_{22} & \cdots & P_{2n}\\ \vdots & \vdots & \ddots & \vdots \\ P_{n1} & P_{n2} & \ldots & P_{nn} \end{array}\right]$$

### 2.2. Markov Transformation Process

For a data series $X=\{x_{1},x_{2},\cdots ,x_{n}\}$, divide it equally into *Q* bins, and each $x_{i}$ in the series has bin $q_{j}\left(j\in \left[1,Q\right]\right)$ corresponding to it; therefore, a Markov matrix *W* of size Q×Q through a first-order Markov chain can be constructed. Its expression is as follows:(8)$$W=\left[\begin{array}{cccc} w_{11} & w_{12} & \cdots & w_{1Q}\\ w_{21} & w_{22} & \cdots & w_{2Q}\\ \vdots & \vdots & \ddots & \vdots \\ w_{Q1} & w_{Q2} & \cdots & w_{QQ} \end{array}\right]$$
(9)$$w_{ij}=p\left\{x_{t}\in q_{i}|x_{t-1}\in q_{j}\right\}$$
where $w_{ij}$ is the probability that the point *x* currently belongs to bin $q_{i}$ will appear in bin $q_{j}$ at the next time. Obviously, $w_{ij}$ satisfies the relation: $\sum_{j=1}^{Q}w_{ij}=1$. Due to the memorylessness of the Markov chain, the current elements in the matrix *W* are independent of the previous element, which results in the loss of much effective information in the process. In order to overcome this shortcoming, by introducing a time axis, the matrix *W* is extended to the Markov transform field *M*. The data set is divided into *Q* bins along the time axis, and the data point at time stamps i and j are belong corresponding bin: $q_{i}$ and $q_{j}$. $M_{ij}$ represents the transition probability of $q_{i}$ to $q_{j}$. It is expressed as below:(10)$$M=\left[\begin{array}{ccc} w_{ij}|x_{1}\in q_{i},x_{1}\in q_{j} & \cdots & w_{ij}|x_{1}\in q_{i},x_{n}\in q_{j}\\ w_{ij}|x_{2}\in q_{i},x_{1}\in q_{j} & \cdots & w_{ij}|x_{2}\in q_{i},x_{n}\in q_{j}\\ \vdots & \ddots & \vdots \\ w_{ij}|x_{n}\in q_{i},x_{1}\in q_{j} & \cdots & w_{ij}|x_{n}\in q_{i},x_{n}\in q_{j} \end{array}\right]$$

After the matrix *M* is obtained, elements in the matrix as pixels, thereby completing the whole process of MTF conversion. The specific process of MTF is shown in Figure 1.

The specific steps are shown below:Divide the vibration signal into *Q* parts.Obtain the probability conversion matrix of Q×Q size.Convert the probability matrix to MTF.Transform MTF into a 2-D image.

## 3. The Brief Introduction of CNN

CNN is a multi-level neural network. Generally speaking, it includes the following structures: convolutional layer, pooling layer and fully connected layer. Next, each type of layer is described below.

### 3.1. Convolutional Layer

The convolution (Conv) layer is named after the convolution operation, but it usually uses the cross-correlation operation between matrices. The Conv layer consists of several convolution units, and the output is generated by the activation unit. The main function of the Conv layer is to extract the hidden features in the input image by calculation.

The convolutional layer has two important hyperparameters: padding (P) and stride (S). Their values affect the size of the output image of the Covn. Padding refers to filling elements in both sides of the height and width of the input matrix (usually filled elements are 0). Padding can increase the width and height of the output image, so that the output and input images have the same width and height. The stride refers to the number of rows and columns of each sliding of the convolution kernel window, which can reduce the width and height of the output image; therefore, after the image passes through the convolution layer, the size of the output picture can be expressed as:(11)$$O=\frac{N-F+2P}{S}+1$$
where *N* and *F* are the sizes of the original input image and convolution kernel, respectively. *P* and *S* are the value of padding and stride. By default, the padding is 0 and the stride is 1.

The activation unit can increase the robustness and nonlinear expression ability of the CNN, which is conducive to better train the model. The common activation functions including Sigmoid, tanh, ReLu and so on. Among them, the ReLu function can improve the running speed of the model and accelerate the convergence of the model. We assume that the convolutional layer *l* uses the convolution kernel *K*, then the entire convolutional layer process is described as follows:(12)$$\begin{array}{cc} y_{l}^{i} & =ReLu(K_{l}^{i}*X_{l-1})\\ & =ReLu(\sum_{j}x_{r-1}^{j}*\omega_{l}^{i}+b_{l}^{i}) \end{array}$$
where $y_{l}^{i}$ means the output of the *i*-th convolution kernel, $\omega_{i}^{l}$ and $b_{i}^{l}$ are the weight and bias of the *i*-th convolution kernel of layer *l*, respectively. $X_{l-1}$ indicates the output of the previous convolutional layer, and its size is *j*. The symbol * represents the operation between the local area and the convolution kernel.

### 3.2. Pooling Layer

The function of the pooling layer is to decrease the size of the feature value space and alleviate the excessive sensitivity of the convolutional layer to position. Compared with the convolutional layer to that calculates the correlation between input and convolutional kernel, The pooling layer mainly keeps the maximum value or average value of elements in the pooling window. The calculation process is also known as maximum pooling or average pooling. In this paper, the operation of pooling layer is maximum pooling, and its expression is as follows:(13)$$H_{l+1}^{n}=\max_{(i-1)r+1\leq t\leq ir}y_{l}^{n}\left(t\right)$$
where, $H_{l+1}^{n}$ represents the output of the maximum pooling layer, *i* is the *i*-th area selected by the pooling window and *r* is the size of the area.

### 3.3. Fully Connected Layer

The fully connected (FC) layer is usually located at the end of the entire CNN. Its main contribution is to flatten the learned feature $H_{l+1}^{n}$ into a vector $V_{m-1}$, and input the obtained vector into the classifier (such as the softmax classifier). It can be expressed as follows:(14)$$\begin{array}{cc} V_{n} & =g(\beta_{n})\\ \beta {}_{n} & ={\left(\delta_{n}\right)}^{T}V_{n-1}+\mu_{n} \end{array}$$
where $\delta_{m}$ and $\mu_{m}$ are the basic parameters of the full connection layer, and *n* is the quantity of them. *g* indicates the nonlinear activation function in the full connection layer.

### 3.4. Proposed CNN Structure

The CNN structure used in this paper is shown in Figure 2, which mainly includes four convolutional layers and four maximum pooling layers. A full connection layer and an output layer (the output layer includes a full connection layer and a Sotfmax classifier). Table 1 shows the specific network structure parameters.

The convolution kernel sizes of the convolutional layers L2, L4, L6 and L8 are all 2×2, the input images of each layer are filled with zero elements. According to the Equation (Equation 11), the size of the output image of each convolutional layer can be calculated. The activation unit used in the convolution layer and pool layer are ReLU. The activation function introduces nonlinearity to the CNN, which is advantageous in solving complex problem. In the FC1 layers, the number of neurons set are 256. The output layer includes a FC layer and a softmax classifier.

Since the size of the filter used in the CNN pooling layer is 2×2, after the image passes through the pooling layer, the output size of the image will be decreased to 1/2 of the original size. The size of input image should be $2^{n}$. Based on this factor and the limitation of the amount of experimental data, the size of the original input image set is 32×32.

## 4. The Process of MTF-CNN

Based on the theory of MTF and CNN, a method called MTF-CNN is proposed, which can accurately and effectively complete the fault classification. The process of MTF-CNN is illustrated in Figure 3, and the specific steps are shown below:Collect the vibration signals.Convert 1-D vibration signals into MTF images.Input the images into CNN for classification.Obtain the result of fault classification.

## 5. Experiment Test

In this part, two data sets were used to verify the feasibility of MTF-CNN. The CNN model was written in Python 3.7 with PyCharm, and the experiment was applied on a computer with a Core I5 7th generation CPU and 8 GB memory.

### 5.1. Experiment 1

#### 5.1.1. Data Description

The data sets used in experiment 1 were obtained from the Bearing Data Center of Case Western Reserve University (CWRU). In this data set, the original vibration signals are obtained by artificially setting fault points in different parts of the rolling bearings, and the experimental platform is illustrated in Figure 4.

The CWRU platform mainly contains three parts. The left end is a two horsepower motor, the middle part is a torque sensor and the right end is a power meter. The electrical discharge machining technology is used to punch out the fault point on the bearing, and its fault range is 0.007 inches to 0.040 inches. We collected a data set containing the normal (NOR) conditions of the rolling bearing and the other three failure conditions (inner race (IR), outer race (OR) and ball (B)). The bearing data parameters used in this paper: sampling frequency is 12 kHz, bearing speed is 1797 rpm, load is 0HP. A total of 1024 points in the vibration signal are taken as a sample, and the sampling time is 0.0853 s. Figure 5 shows the original vibration images and corresponding spectra in the CUWR dataset under four different working conditions.

#### 5.1.2. Result Analysis

The vibration signals in the four cases are converted into MTF, and the conversion results are shown in Figure 6. It can be seen that under different fault conditions, these converted images are totally different from each other, and these images can be intuitively classified.

We collected 100 samples under each working condition, for a total of 400 samples. The algorithm randomly selects 70% of the total data sets as the training set and the rest 30% as the test set. A total of 10 rounds of training were carried out, and the number of epochs of each training is 100. The purpose of this approach is to avoid the special data partition, which leads to the deviation of classification results.

Since the training sets and test sets of the CNN model in each round of training are random, the classification results of each round will be different. In the course of ten rounds of training, the highest classification accuracy is 100%, and the lowest is 99.18%. The average accuracy rate is 99.88%. The confusion matrices of the CNN classification result are shown in Figure 7.

In order to show the advantages of MTF-CNN in classification accuracy, other deep learning methods were selected for comparison under the same conditions. The details are shown in Table 2. The LSTM model is a time recursive network. The classifier used for data processing is softmax and the controller is Adam. The learning rate of the LSTM model used in this article is 0.001. The batch-size is 27 and the epochs are 100. SVM is a learning model that can be used for classification. The SVM in this paper uses GridSearchCV (5−fold cross verification parameters). The kernel function uses a radial basis kernel function (RBF). The feature values of LSTM and SVM are directly composed of points in the data sets. At the same time, the fuzzy entropy (FE) is introduced as the feature value of the two classification algorithms to compare the classification effects.

From the Table 2, we can see that the MTF-CNN is far better than LSTM and SVM in terms of classification effect. Among them, the average accuracy rate of MTF-CNN is about 99.88%. For the classification effect of LSTM and SVM, it is better to use fuzzy entropy as the feature value than to directly use the points in the data set as the feature value. Moreover, during the ten experiments of MTF-CNN, the fluctuation of classification accuracy is smaller than that of the other two methods.

In [30], the author analyzes the application of the CWRU data set in bearing fault diagnosis. According to [30], the data are divided into three categories: diagnosable (category Y), partially diagnosable (category P) and non-diagnosable (category N). At the same time, the literature holds that the new fault diagnosis method should be verified by P-category or N-category data sets. The normal data and outer ring data used in this paper belong to the Y-category, and the ball data belongs to the N-category; therefore, the data set used in this paper leads to similar conclusions.

### 5.2. Experiment 2

#### 5.2.1. Data Description

This data set comes from Mechanical Failure Prevention Technology (MFPT), including the normal (NOR) condition of the fixed load, outer ring (OR) failure and inner ring (IR) failure under various loads. The parameters of the normal data set used in this paper are as follows: the load is 270 lbs and the input shaft speed is 25 Hz. Outer and inner fault data parameters: load 300 lbs and input shaft speed 25 Hz. A total of 1024 vibration signal data points were taken as a sample. The original vibration images and corresponding spectra in the MFPT dataset under three different working conditions are shown in Figure 8.

#### 5.2.2. Result Analysis

Convert different types of vibration signals into the corresponding MTF images and the conversion results are shown in Figure 9. It can be seen from the figure that there are obvious differences in pixel distribution among different types of MTF images.

A total of 100 samples were collected under each working condition, for a total of 300. The data set division is the same as experiment 1, and the train set and test set account for 70% and 30% of the total number, respectively. A total of ten rounds of experiments were conducted, and each round was run for 100 epochs. The purpose of this practice is to avoid biased results due to special training and test sets. In ten rounds of experiments, the CNN classification accuracy under this data set has reached 100%, and the classification results do not have obvious fluctuations for different test sets. Its confusion matrix is shown in Figure 10.

Same as experiment 1, the classification accuracy of MTF-CNN and other methods is compared under MPTF data set, and the detailed comparison results are shown in Table 3. As can be seen from the table, the classification accuracy of MTF-CNN is higher than that of other methods. At the same time, the accuracy rate did not fluctuate greatly in multiple experiments, which shows that MTF-CNN has better robustness.

In summary, two experiments prove that MTF-CNN can accurately and effectively identify pictures with different labels, thus realizing the classification and diagnosis of faults. Meanwhile, compared with other methods such as LSTM, MTF-CNN is superior in classification accuracy.

## 6. Conclusions

In this paper, a method for diagnosing rolling bearing faults based on MTF-CNN is proposed. It eliminates the tedious signal processing work that is required to extract the fault information from the original data and reduces the manpower cost due to its autonomous classification. Furthermore, compared to the other popular machine learning algorithms, the proposed method produces better accuracy with reduced number of data samples. The effectiveness of the MTF-CNN is verified by experimental results. The number of data samples used in this paper is less than that of general papers; however, by optimizing the structure of CNN, the dependence of CNN model on training samples can be reduced.

## Figures and Tables

**Figure 1 entropy-24-00751-f001:**
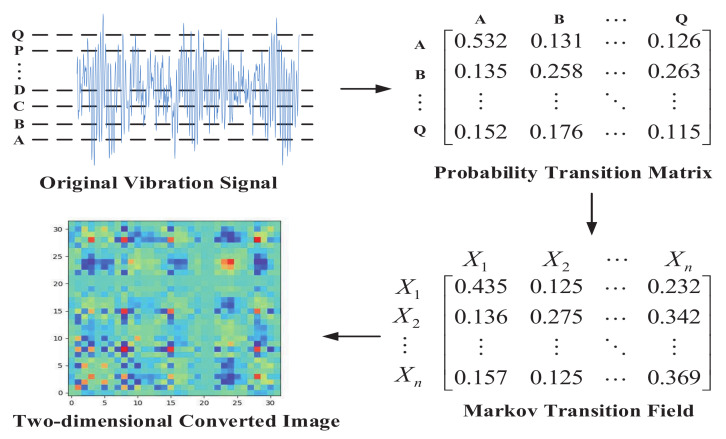
The flow chart of the MTF.

**Figure 2 entropy-24-00751-f002:**
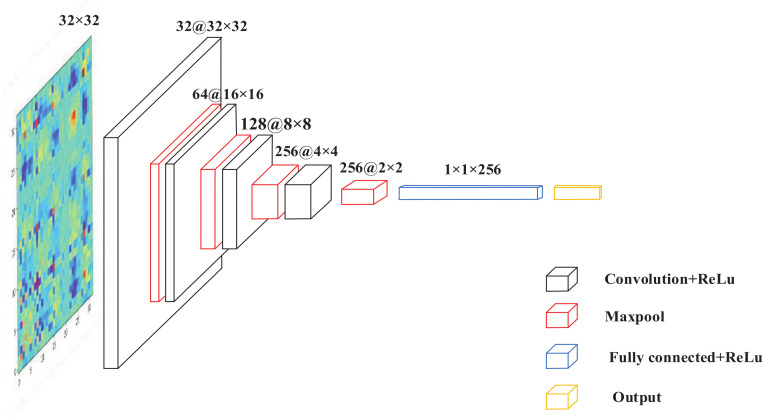
CNN model structure.

**Figure 3 entropy-24-00751-f003:**
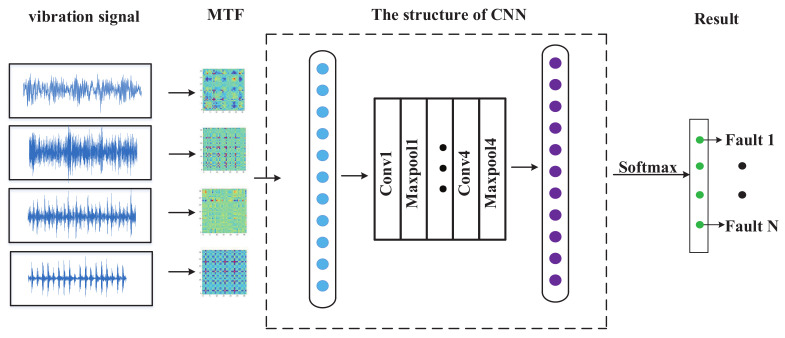
The flow chart of the MTF-CNN.

**Figure 4 entropy-24-00751-f004:**
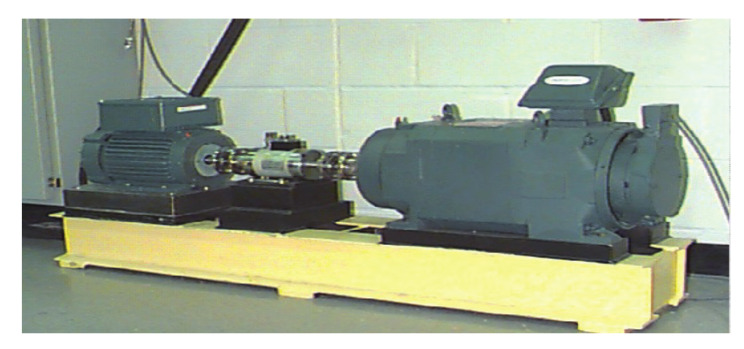
CWRU bearing platform.

**Figure 5 entropy-24-00751-f005:**
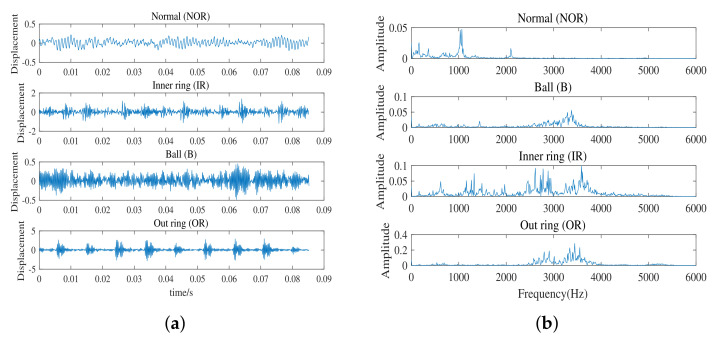
Waveforms of vibration signal in four conditions. (**a**) The diagram of time domain. (**b**) The diagram of frequency domain.

**Figure 6 entropy-24-00751-f006:**
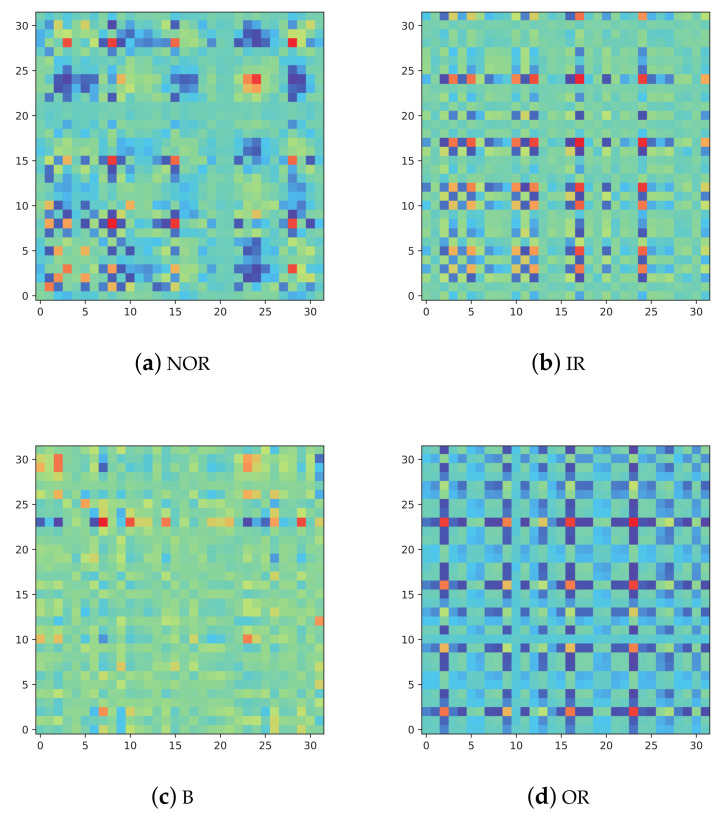
The converted images on four fault conditions.

**Figure 7 entropy-24-00751-f007:**
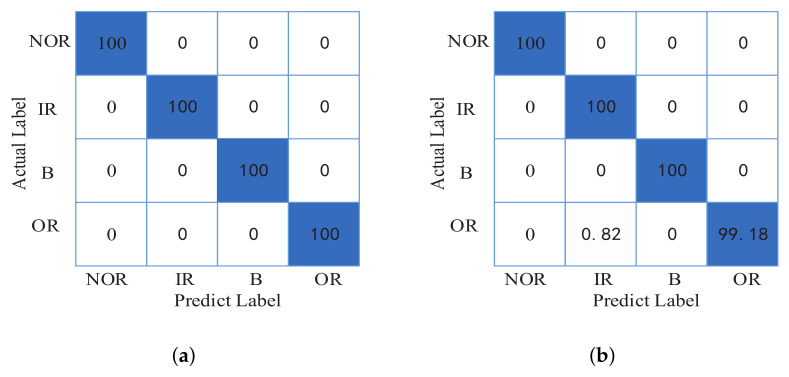
The confusion matrix of CNN in CWRU data set. (**a**) The highest classification result. (**b**) The lowest classification result.

**Figure 8 entropy-24-00751-f008:**
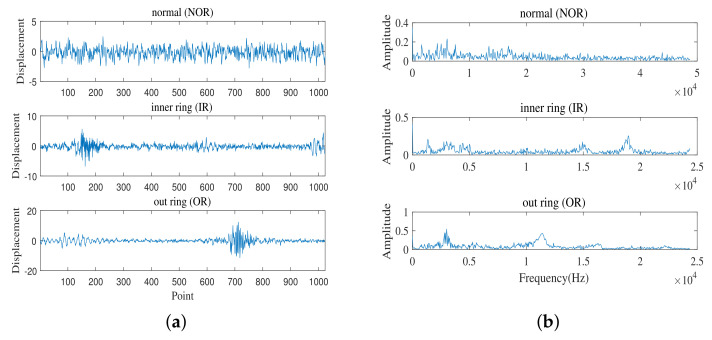
Waveforms of vibration signal in three conditions. (**a**) The diagram of time domain. (**b**) The diagram of frequency domain.

**Figure 9 entropy-24-00751-f009:**
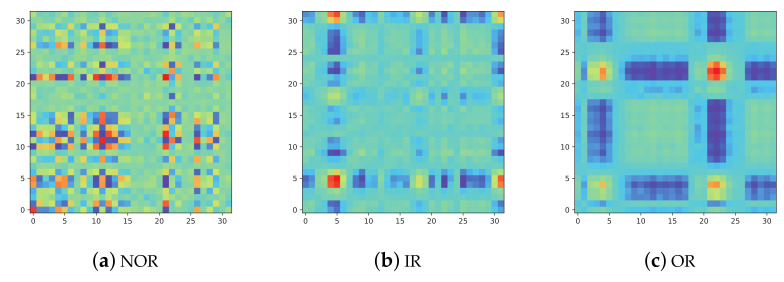
The converted images on three fault conditions.

**Figure 10 entropy-24-00751-f010:**
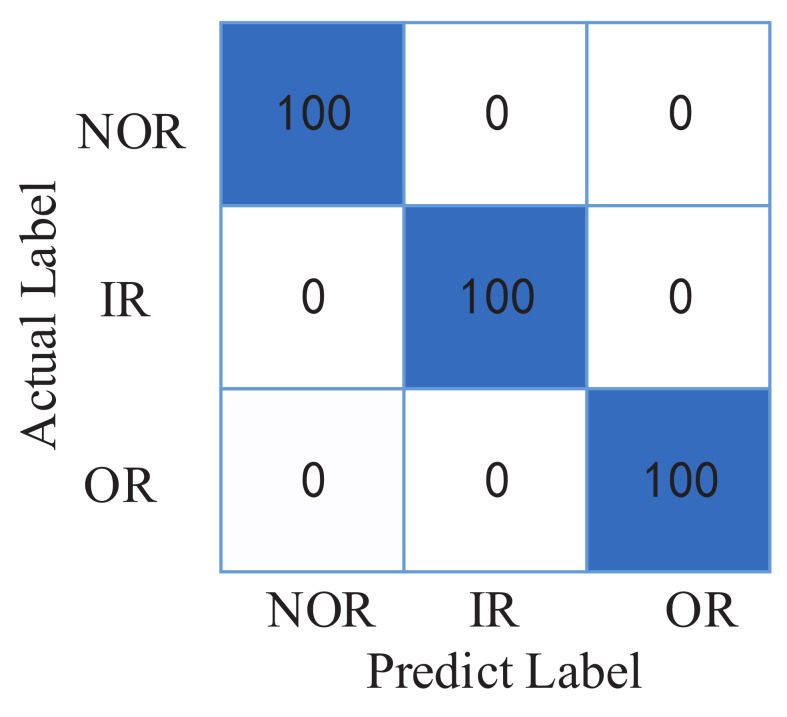
The confusion matrix of CNN in MFPT data set.

**Table 1 entropy-24-00751-t001:** The parameters of CNN.

Layer	CNN Models	Kernel Size	Padding	Stride
Layer1	Conv	3×3	1	1
Layer2	Maxpool	2×2	No	1
Layer3	Conv	3×3	1	1
Layer4	Maxpool	2×2	No	1
Layer5	Conv	3×3	1	1
Layer6	Maxpool	2×2	No	1
Layer7	Conv	3×3	1	1
Layer8	Maxpool	2×2	No	1
Layer9	FCl	256	-	-

**Table 2 entropy-24-00751-t002:** The accuracy of six methods.

Method	Highest Accuracy	Lowest Accuracy	Mean
MTF-CNN	100%	99.18%	99.88%
LSTM	80.4%	60.3%	73.8%
FE-LSTM	91.4%	88.7%	90.3%
SVM	90.3%	83.5%	89.4%
FE-SVM	98.75%	96.25	97.5%

**Table 3 entropy-24-00751-t003:** The accuracy of six methods.

Method	Highest Accuracy	Lowest Accuracy	Mean
MTF-CNN	100%	100%	100%
LSTM	87.6%	71.5%	79.8%
FE-LSTM	95.4%	90.7%	93.3%
SVM	92.6%	88.5%	91.5%
FE-SVM	99%	97%	98.6%

## Data Availability

The experimental data are available from the Bearing Data Center of Case Western Reserve University deposited in https://engineering.case.edu/bearingdatacenter/welcome (accessed on 1 May 2022) and the Mechanical Failure Prevention Technology deposited in https://www.mfpt.org/fault-data-sets/(accessed on 1 May 2022).

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
