# Peer review of "A New Fault Diagnosis of Rolling Bearing Based on Markov Transition Field and CNN"

_entropy, 2022, doi:10.3390/e24060751_

Round 1
Reviewer 1 Report
Manuscript ID: Entropy-1734255
Manuscript Title: A new fault diagnosis of rolling bearing based on Markov transition field and CNN
The authors investigated a new fault diagnosis technique based on the Markov transition field and CNN. This article clearly belongs to this journal. The presentation of the manuscript looks great. The proposed Methodology is clear. The Results are very well presented, and the conclusion is clear and concise. However, it has a few issues with grammar and spelling. The authors should review the whole manuscript to improve the manuscript. Moreover, the authors may highlight some of the important results of the study in the abstract section to attract the reader and to make it a more interesting article. Furthermore, the author should provide clear Figure 4 on page 7. The provided figure looks stressed out from the original image.
Author Response
Reviewer 1:
The authors investigated a new fault diagnosis technique based on the Markov transition field and CNN. This article clearly belongs to this journal. The presentation of the manuscript looks great. The proposed Methodology is clear. The Results are very well presented, and the conclusion is clear and concise. However, it has a few issues with grammar and spelling. The authors should review the whole manuscript to improve the manuscript. Moreover, the authors may highlight some of the important results of the study in the abstract section to attract the reader and to make it a more interesting article. Furthermore, the author should provide clear Figure 4 on page 7. The provided figure looks stressed out from the original image.
Our Response: In response to your comments on the grammar and spelling of the manuscript, we have re-reviewed the manuscript and modified the questions accordingly. At the same time, we have rewritten the Abstract section of the manuscript to emphasize the findings of the manuscript. Figure 4 shows the public dataset CWRU collection platform. We can only improve the clarity of the picture as much as possible to meet the requirements.
Reviewer 2 Report
The article deals with a new approach to vibration diagnostics of rotary machines. As a result of the comprehensive application of convolutional neural networks and Markov chains, the proposed approach increases classification accuracy and robustness.
However, despite the undoubtful practical significance of the submitted manuscript in fault diagnosis of rolling bearings and ensuring vibration reliability of rotary machines, the following minor corrections should be eliminated before publication:
- Since the article aims at fault diagnosis of rolling bearings, the state-of-the-art studies in different practical applications should be reflected in the Introduction chapter, e.g., condition monitoring of turbine bearings using vibration diagnostics, etc.
- Since the MTF and CNN are well-known approaches, please specify a particular contribution of the authors in these fields.
- The corresponding frequency responses should supplement waveforms of vibration signals (Figures 5 and 8).
- In Figures 5 and 8, maybe “Amplitude” should be renamed as “Displacement”?
- Figures 7 and 10 can be merged.
Author Response
Reviewer 2:
The article deals with a new approach to vibration diagnostics of rotary machines. As a result of the comprehensive application of convolutional neural networks and Markov chains, the proposed approach increases classification accuracy and robustness. However, despite the undoubtful practical significance of the submitted manuscript in fault diagnosis of rolling bearings and ensuring vibration reliability of rotary machines, the following minor corrections should be eliminated before publication.
Our Response: Thanks for your guidance and suggestions for this manuscript. Regarding your specific comments on the manuscript, our responses are as follows.
Point 1: ("Since the article aims at fault diagnosis of rolling bearings, the state-of-the-art studies in different practical applications should be reflected in the Introduction chapter, e.g., condition monitoring of turbine bearings using vibration diagnostics, etc.")
Our Response 1: Thanks for your comments. We have updated our list of references in the revised manuscript.
Point 2: ("Since the MTF and CNN are well-known approaches, please specify a particular contribution of the authors in these fields.")
Our Response 2: Thank you for your careful review. W Aiming at the problem of converting one-dimensional time series into two-dimensional images, Markov domain theory (MTF) is introduced to realize the data conversion. Based on MTF and convolutional neural network (CNN), a fault diagnosis method (MTF-CNN) is proposed. A CNN model for fault diagnosis of rolling bearings is proposed by setting the structure of the model and the number of neurons. The data set is used to verify MTF-CNN, and the experimental results indicate that this method can effectively classify faults, and has certain advantages compared with the same type of methods. Aiming at the problem of converting one-dimensional time series into two-dimensional images, Markov domain theory (MTF) is introduced to realize the data conversion. Based on MTF and convolutional neural network (CNN), a fault diagnosis method (MTF-CNN) is proposed. By setting the structure of the model and the number of neurons, a CNN model for fault diagnosis of rolling bearings is proposed. The data set is used to verify MTF-CNN, and the experimental results indicate that this method can effectively classify faults, and has certain advantages compared with the same type of methods.
Point 3: ("The corresponding frequency responses should supplement waveforms of vibration signals (Figures 5 and 8).")
Our Response 3: Thanks for your valuable comment and we appreciate it. We have supplemented the corresponding frequency waveforms of vibration signals. The modified images are on page 8, figure 5 and page 10, figure 8 in the revised manuscript.
Point 4: ("In Figures 5 and 8, maybe “Amplitude” should be renamed as “Displacement”? ")
Our Response 4: Thank you for your positive suggestion on this manuscript. We have renamed the "Amplitude" in Figure 5 and Figure 8 as "Displacements".
Point 5: ("Figures 7 and 10 can be merged. ")
Our Response 5: Thanks for your valuable comment. We apologize for not clarifying this part. Figure 7 shows the classification accuracy of MTF-CNN in CWRU data set, and Figure 10 shows the classification results of MTFP data. The contents of the two pictures are different, so they are presented separately in the manuscript. In order to clear up the misunderstanding, we have modified the captions of Figure 7 and Figure 10 accordingly.